# Crystal Structure of Pyrrolysyl-tRNA Synthetase from a Methanogenic Archaeon ISO4-G1 and Its Structure-Based Engineering for Highly-Productive Cell-Free Genetic Code Expansion with Non-Canonical Amino Acids

**DOI:** 10.3390/ijms24076256

**Published:** 2023-03-26

**Authors:** Tatsuo Yanagisawa, Eiko Seki, Hiroaki Tanabe, Yoshifumi Fujii, Kensaku Sakamoto, Shigeyuki Yokoyama

**Affiliations:** 1RIKEN Center for Biosystems Dynamics Research, 1-7-22 Suehiro-Cho, Tsurumi, Yokohama 230-0045, Japan; kensaku.sakamoto@riken.jp; 2RIKEN Cluster for Science, Technology and Innovation Hub, 1-7-22 Suehiro-Cho, Tsurumi, Yokohama 230-0045, Japan; eiko.seki@riken.jp (E.S.); hiroaki.tanabe@riken.jp (H.T.);

**Keywords:** non-canonical amino acids, genetic code expansion, crystal structure, tRNA, cell-free protein synthesis

## Abstract

Pairs of pyrrolysyl-tRNA synthetase (PylRS) and tRNA^Pyl^ from *Methanosarcina mazei* and *Methanosarcina barkeri* are widely used for site-specific incorporations of non-canonical amino acids into proteins (genetic code expansion). Previously, we achieved full productivity of cell-free protein synthesis for bulky non-canonical amino acids, including *N*^ε^-((((*E*)-cyclooct-2-en-1-yl)oxy)carbonyl)-L-lysine (TCO*Lys), by using *Methanomethylophilus alvus* PylRS with structure-based mutations in and around the amino acid binding pocket (first-layer and second-layer mutations, respectively). Recently, the PylRS·tRNA^Pyl^ pair from a methanogenic archaeon ISO4-G1 was used for genetic code expansion. In the present study, we determined the crystal structure of the methanogenic archaeon ISO4-G1 PylRS (ISO4-G1 PylRS) and compared it with those of structure-known PylRSs. Based on the ISO4-G1 PylRS structure, we attempted the site-specific incorporation of *N*^ε^-(*p*-ethynylbenzyloxycarbonyl)-L-lysine (*p*EtZLys) into proteins, but it was much less efficient than that of TCO*Lys with *M. alvus* PylRS mutants. Thus, the first-layer mutations (Y125A and M128L) of ISO4-G1 PylRS, with no additional second-layer mutations, increased the protein productivity with *p*EtZLys up to 57 ± 8% of that with TCO*Lys at high enzyme concentrations in the cell-free protein synthesis.

## 1. Introduction

Expanding the genetic code with non-canonical amino acids is useful for developing novel structures and functions of proteins (reviewed in [1,2]). Site-specific incorporation of non-canonical amino acids into proteins in response to specified (e.g., UAG) codons has been achieved by pairs of an engineered aminoacyl-tRNA synthetase (aaRS) and tRNA, including the pairs of pyrrolysyl-tRNA synthetase (PylRS) and tRNA^Pyl^ _(CUA)_ [2,3,4,5,6,7,8,9,10]. The PylRS·tRNA^Pyl^ pair was first found in methanogenic archaea, including *Methanosarcina barkeri* [11,12], and in bacteria, including *Desulfitobacterium hafniense* [13,14]. The PylRS·tRNA^Pyl^ pairs from *M. barkeri* and *Methanosarcina mazei* have been extensively studied (reviewed in [10,15,16,17,18,19,20,21]). Recently, by using the pairs of PylRS and tRNA^Pyl^ from *Methanomethylophilus alvus*, the methanogenic archaeon ISO4-G1, the methanogenic archaeon ISO4-H5, *Methanomassiliicoccus intestinalis*, and *Methanomassiliicoccus luminyensis*, the site-specific incorporations of non-canonical amino acids into proteins have been achieved [22,23,24,25,26,27,28,29,30,31,32,33].

The PylRS·tRNA^Pyl^ pairs are useful for non-canonical amino acid incorporation because of their “orthogonality” (non-reactivity) to the 20 canonical aaRS·tRNA pairs in many organisms [15,16,17,18,19]. PylRS and its mutants showed broad specificity for substrate amino acids, and by using the PylRS·tRNA^Pyl^ pairs, site-specific incorporations of more than 200 non-canonical amino acids into proteins have been achieved in bacteria including *Escherichia coli*, and eukaryotes including *Saccharomyces cerevisiae*, mammalian cells, and multicellular organisms (reviewed in [10,15,16,17,18,19,20,21,34]), and by cell-free protein synthesis based on an *E. coli* cell extract [28,35,36,37,38,39,40,41,42]. Cell-free protein synthesis systems, which are novel protein expression platforms, are particularly suitable for synthesizing cell-toxic proteins and transmembrane proteins that are difficult to synthesize in cellular systems, and can efficiently introduce non-canonical amino acids into such proteins for pharmaceutical research.

The efficiencies of non-canonical amino acid incorporations into proteins at the UAG codon are usually lower than that of the standard protein synthesis with a canonical amino acid at the corresponding position. The incorporation efficiency of a non-canonical amino acid at multiple sites in a protein is much lower than that of the single-site incorporation. Therefore, by using *E. coli* strains lacking translation termination factor 1 (RF-1) to achieve the complete reassignment of the UAG codon [35,43,44,45,46,47,48], we have increased the incorporation efficiencies of non-canonical amino acids to the maximum level, which was previously designated as the full productivity of the expanded genetic code [26].

*M. mazei* PylRS (*Mm*PylRS) and *M. barkeri* PylRS (*Mb*PylRS) consist of the N- and C-terminal domains (PylRSn and PylRSc, respectively). Another group of PylRSs from bacteria, including *D. hafniense*, is composed of two separate gene products (PylSn and PylSc), which are homologous to PylRSn and PylRSc, respectively [12,13,14,49]. The PylRSc protein exhibited higher solubility than the full-length PylRS protein and was easily crystallized [50]. However, the PylRSc protein retained insufficient tRNA binding and aminoacylation activities [51,52]. Consequently, both the N- and C-terminal domains of PylRS (i.e., full-length PylRS) have been regarded as essential components for the efficient incorporation of non-canonical amino acids into proteins. Notably, recently discovered methanogenic archaea, including *M. alvus*, *M. intestinalis*, *M. luminyensis*, ISO4-G1, and ISO4-H5, have PylSc homologs, but lack the genes encoding PylSn homologs in their genomes [22,53]. The high solubility of the PylSc-type *M. alvus* PylRS makes it suitable for crystallographic analysis and cell-free protein synthesis for non-canonical amino acid incorporation [26].

Crystal structures of PylRSs from *M. mazei* and *D. hafniense* and their complexes have been extensively investigated [13,14,49,52,54]. The catalytic fragment (residues 185–454) of *M. mazei* PylRS (*Mm*PylRSc) has been crystallized [50], and the structures of *Mm*PylRSc and its mutants in complex with numerous substrate amino acids, aminoacyladenylates, and ATP and its analogs have been determined [8,50,52,54,55,56,57,58,59,60,61,62,63,64,65]. Based on the structural information and random screening, we obtained *Mm*PylRS with the Y306A and Y384F mutations [*Mm*PylRS (Y306A/Y384F)] [8]. The pair of *Mm*PylRS (Y306A/Y384F) and tRNA^Pyl^ is one of the useful aaRS·tRNA pairs for cellular and cell-free genetic code expansion with bulky non-natural lysine derivatives, including *N*^ε^-benzyloxycarbonyl-L-lysine (ZLys), *N*^ε^-(*o*-azidobenzyloxycarbonyl)-L-lysine (*o*AzZLys), and *N*^ε^-(*m*-azidobenzyloxycarbonyl)-L-lysine (*m*AzZLys). A variety of applications using the pair have been developed [8,26,59,66,67,68,69,70,71,72,73,74]. The crystal structures of the catalytic fragment of *Mm*PylRS (Y306A/Y384F) [*Mm*PylRSc (Y306A/Y384F)] in complex with 14 bulky non-natural lysine derivatives, including ZLys, *m*AzZLys, and *N*^ε^-((((*E*)-cyclooct-2-en-1-yl)oxy)carbonyl)-L-lysine (TCO*Lys), revealed the structural bases for their amino acid binding modes [63].

We recently determined the crystal structure of *M. alvus* PylRS (*Ma*PylRS) [26]. The *Ma*PylRS with the Y126A (corresponding to the Y306A mutation in *M. mazei* PylRS) and M129L mutations efficiently incorporated bulky non-canonical amino acids, including ZLys and *m*AzZLys, into proteins in *E. coli* cells [24,26]. Tyr126 and Met129 are the “first-layer residues”, located within ~5 Å from (and in direct contact with) substrate amino acids; while, His227 and Tyr228 are the “second-layer residues”, which are located within ~9 Å from (and do not directly contact) substrates but might affect the first-layer residues. This resembles the concept of first-shell and second-shell residues, respectively, as reported by others [75]. The *Ma*PylRS with both the first-layer (Y126A and M129L) and second-layer (H227I and Y228P) mutations enhanced the protein productivities drastically to the maximum level (i.e., full productivities) for ZLys, *m*AzZLys, and the more difficult amino acid TCO*Lys by *E. coli* cell-free protein synthesis, as compared with those obtained with the *Mm*PylRS (Y306A/Y384F) and *Ma*PylRS (Y126A/M129L) pairs [26]. Recently, the crystal structures of *Ma*PylRS (N166S/V168C/W239T) and *Ma*PylRS (N166A/C168G/W239C) mutants in complex with the fluorescent non-canonical amino acid acrydonylalanine and ATP (or its non-hydrolyzable analog, adenylyl imidodiphosphate (AMPPNP)) were determined, and the conformational changes of the residues including His227 upon non-canonical amino acid-binding were discussed [76]. It should be emphasized that *Ma*PylRS is much more useful than *Mm*PylRS for cell-free genetic code expansion with bulky non-canonical amino acids [26]. However, some bulky non-canonical amino acids still cannot be incorporated by using the *Ma*PylRS mutant system. To synthesize proteins in which these bulky non-canonical amino acids are efficiently incorporated, we should compare the structures of PylRSs from archaea, and create active mutants for inefficient bulky non-canonical amino acids based on these structures.

The PylSc-type PylRS from the methanogenic archaeon ISO4-G1 (ISO4-G1 PylRS) is highly similar to *Ma*PylRS with 64% sequence identity, while ISO4-G1 PylRS and *Ma*PylRS both share around 38% sequence identity with *M. mazei* PylRSc. However, the pair of PylRS and tRNA^Pyl^ from the methanogenic archaeon ISO4-G1 (ISO4-G1 PylRS·tRNA^Pyl^) is orthogonal to the *M. mazei* pair, but the *M. alvus* pair is not [22,23,27]. The mechanism by which ISO4-G1 PylRS recognizes tRNA^Pyl^ and a variety of non-canonical amino acids in its active site must be elucidated for understanding its substrate specificity and orthogonality, and for achieving superior genetic code expansion systems. Recently, the crystal structure of ISO4-G1 PylRS with the L124A, Y125L, V167A, Y204W, and A221S mutations for the incorporation of the non-canonical amino acid *m*-cyanopyridylalanine was determined in the apo form [32].

In the present study, we solved the crystal structure of the wild-type ISO4-G1 PylRS apo form, and compared it with that of the multiple mutant of ISO4-G1 PylRS [32]. In the structure of the wild-type ISO4-G1 PylRS, His225 appears to be located in a different position from that in the multiple mutant ISO4-G1 PylRS structure. In this context, in *Ma*PylRS, the conformational changes of the corresponding His227 residue are considered to be important for the PylRS activity. Therefore, the structural changes of the conserved His residues may be common features between the two PylRSs, and might be the driving force for the movement of the specific hairpin (the β5-β6 hairpin, which will be described later) and thus crucial for the PylRS activity. Furthermore, in the present study, the ISO4-G1 PylRS mutants engineered based on the ISO4-G1 PylRS structures have been applied for the milligram-scale preparation of proteins containing useful non-canonical amino acids, including TCO*Lys and *N*^ε^-(*p*-ethynylbenzyloxycarbonyl)-L-lysine (*p*EtZLys), in the cell-free protein synthesis system. These rationally engineered ISO4-G1 PylRS mutants will be more useful than ever before for genetic encode expansion technologies.

## 2. Results

### 2.1. Overall Structure of ISO4-G1 PylRS

The genome of the methanogenic archaeon ISO4-G1 encodes an ISO4-G1 PylRS protein consisting of 273 amino acids [53], which is quite similar to *Ma*PylRS [22,53] (Appendix A). ISO4-G1 PylRS was expressed very well as a soluble protein in *Escherichia coli* cells. The yield of the ISO4-G1 PylRS protein was over 100 mg per liter *E. coli* culture (Appendix A), and the protein could be concentrated without aggregation to more than 20 mg/mL, which is comparable to that of *M. alvus* PylRS. For crystallographic analysis, ISO4-G1 PylRS was purified to homogeneity by three column chromatography steps (Materials and Methods). The crystallization of the ISO4-G1 PylRS protein was successful when using PEG3350 as the precipitant, and the crystal structure of the ISO4-G1 PylRS apo form has been determined at 2.78-Å resolution (Figure 1, Materials and Methods). The structure of the ISO4-G1 PylRS protein is shown in Figure 1. The asymmetric unit contains ten molecules of PylRS (five PylRS dimers, A/B, C/G, D/F, E/H, J/I). The final model shows good geometry, and all residues are within the allowed regions of the Ramachandran plot, as evaluated by Procheck [77] and Molprobity [78] (Appendix A).

### 2.2. Structure-Based Sequence Comparison of ISO4-G1 PylRS with Other PylRSs

In the ISO4-G1 PylRS structure, the residues from Met1 to Pro58 contain two α-helices (α1 and α2) (Figure 1b and Appendix A). The following residues Asn60-Asp273 constitute the catalytic domain, with the characteristic topology of the class-II aaRSs: an extended six-strand, anti-parallel β-sheet (β3, β4, β5, β6, β7, β8) surrounded by α-helices. The sequence motifs (motifs 1, 2, and 3), which are conserved in class-II aaRSs [79,80,81], correspond to the residues Gly78-Val88, Cys147-Leu164, and Ala238-Lys251, respectively, in ISO4-G1 PylRS (Figure 1b and Appendix A). The structures of the ordering loop (residues Ile97-Gln106) and the motif-2 loop (residues Lys150-Glu159) are quite similar in the ten PylRS molecules, while the β5-β6 hairpin (residues Thr195-Ile212) adopts open and closed conformations (Figure 1 and Appendix A). The β5-β6 hairpin in ISO4-G1 PylRS corresponds to the β5-β6 hairpin in *Ma*PylRS and the β7-β8 hairpin in *Mm*PylRSc, which randomly adopt open and closed conformations as described previously [26,52].

### 2.3. Structural Comparison of ISO4-G1 PylRS with MaPylRS, DhPylSc, and MmPylRSc

The structure of ISO4-G1 PylRS is highly homologous with those of *Ma*PylRS, *Dh*PylSc, and *Mm*PylRSc (Figure 2). A DALI search (http://www.embl-ebi.ac.uk/dali (accessed on 8 December 2022).) revealed that the ISO4-G1 PylRS structure (molecule B) resembles those of the PylRS proteins (PDBs: 6JP2, 6EZD, 7R6O, 2ZNI, 2ZNJ, 3DSQ, 2E3C, 4CH3, 2ZIN, and 2ZIM), with Z-scores of 31.4–33.7, 30.0–30.6, 33.0–33.3, 30.5–30.7, 29.6–30.9, 30.2, 28.9, 28.6, 28.4, and 27.6, respectively. Residues Asp61–Asp273 of ISO4-G1 PylRS constitute the catalytic core and are superimposed well on *Ma*PylRS, *Dh*PylSc (residues Ala72–Asn286), and *Mm*PylRSc (residues Tyr242–Asn453), but the two α-helices (α1 and α2, residues Met1–Ala56) in the N-terminal 59 residues of ISO4-G1 PylRS are slightly different from those of the other PylRSs. The superimposition of the ISO4-G1 PylRS structure on those of *Ma*PylRS (PDB: 6JP2), *Mm*PylRSc (PDB: 2ZIM), and *Dh*PylSc·tRNA^Pyl^ (PDB: 2ZNI) revealed that the N-terminal two α-helices of ISO4-G1 PylRS and *Mm*PylRSc cause steric hindrance with tRNA^Pyl^, in contrast to those of *Ma*PylRS and *Dh*PylSc (Figure 2 and Appendix A). Accordingly, the two α-helices of PylRSs might undergo conformational changes upon tRNA^Pyl^ binding.

### 2.4. Structural Comparison of the Amino Acid Binding Residues of ISO4-G1 PylRS with Those of MaPylRS and MmPylRSc

Based on the structure-based sequence alignments of PylRSs (Appendix A), Ala121, Leu124, Tyr125, Asn165, Leu227, and Trp237 in the amino acid binding pocket of ISO4-G1 PylRS are conserved among *Ma*PylRS (Ala122, Leu125, Tyr126, Asn166, Leu229, and Trp239, respectively) and *Mm*PylRSc (Ala302, Leu305, Tyr306, Asn346, Leu407, and Trp417, respectively), whereas the counterparts of Met128, Val167, and Ala221 in ISO4-G1 PylRS correspond to Leu309, Cys348, and Val401, respectively, in *Mm*PylRSc. The present crystallographic analysis revealed the structural differences between ISO4-G1 PylRS, *Ma*PylRS, and *Mm*PylRSc (PDB: 2ZIM) (Figure 3). The ISO4-G1 PylRS Met128 and Val167 residues, which are, respectively, conserved as Met129 and Val168 in *Ma*PylRS, are bulkier than the corresponding Leu309 and Cys348 residues in *Mm*PylRSc, respectively. Therefore, the internal volumes of the amino acid binding pockets of ISO4-G1 PylRS and *Ma*PylRS are smaller than that of *Mm*PylRSc.

### 2.5. Structural Changes of Tyr204 and His225 in ISO4-G1 PylRS and Comparison with Those of MmPylRS and MaPylRS

Tyr204, at the tip of the β5-β6 hairpin in ISO4-G1 PylRS molecules A, C, E, F, and J, is located far from the amino acid binding pocket (the open conformation), while in ISO4-G1 PylRS molecules B, D, G, H, and I, Tyr204 is located inside the amino acid binding pocket (the closed conformation) (Figure 2, Figure 4, Figure 5 and Appendix A).

Therefore, the β5-β6 hairpin undergoes random conformational changes. The Tyr204 side-chain in molecule D is disordered and thus might be in an intermediate form (Figure 5i). The β5-β6 hairpin in ISO4-G1 PylRS is similar to that of *Ma*PylRS, while a remarkable difference exists between the β5-β6 hairpin in ISO4-G1 PylRS and the β7-β8 hairpin in *Mm*PylRSc. As described previously, the β7-β8 hairpin in *Mm*PylRSc is very flexible and adopts multiple conformations regardless of the substrate binding [52] (Figure 2 and Figure 4). In the *Mm*PylRSc structure (PDB: 2ZIM), the β7-β8 hairpin is bent in the middle, and the tip half of the β-hairpin is elevated. Tyr384, at the tip of the bent β7-β8 hairpin, is buried deeply within the active site [52,54]. While the β5-β6 hairpin in ISO4-G1 PylRS still assumes bent conformations, Tyr204 of ISO4-G1 PylRS is not as deeply accommodated within the active site as compared to Tyr384 of *Mm*PylRSc (Figure 2, Figure 3 and Figure 4a,b,h). In the crystal structure of the ISO4-G1 PylRS mutant for cyanopyridylalanine [32], Trp204, which is substituted for the strictly conserved Tyr residue in the PylRS family, at the tip of the bent β5-β6 hairpin is only slightly inserted into the active site as compared to Tyr204 of the ISO4-G1 PylRS apo form (Figure 4c and Appendix A). In contrast, in the crystal structures of the acrydonylalanine (and ATP/AMPPNP)-bound *Ma*PylRS mutants [76], Tyr206 at the tip of the bent β5-β6 hairpin seems to penetrate more deeply within the active site than Tyr206 in the apo form of *Ma*PylRS (Figure 4f and Appendix A). The Tyr204/Trp204 residues of the ISO4-G1 PylRSs in the apo form are shallowly inserted within their active sites, as compared to Tyr206 of the *Ma*PylRS mutant. Accordingly, the two structures of ISO4-G1 PylRS are ligand-free forms, in which Tyr204/Trp204 adopt open and partially closed conformations (Figure 4a–c and Appendix A); whereas, the *Ma*PylRS mutant structure represents the amino acid substrate and ATP/AMPPNP-bound form, and Tyr206 adopts the completely closed conformation (Figure 4f,h and Appendix A). Notably, Tyr206 of the AMPPNP-bound form of the *Ma*PylRS mutant was completely disordered (Appendix A). Consequently, in the case of *Ma*PylRS, the bound amino acid substrate may induce Tyr206 at the tip of the β5-β6 hairpin to adopt the closed conformation.

Interestingly, the ISO4-G1 PylRS structure revealed that the His225 residue (corresponding to His227 in *Ma*PylRS) undergoes drastic conformational changes in accordance with the location of the Tyr204 residue (corresponding to Tyr206 in *Ma*PylRS) in the β5-β6 hairpin (Figure 2, Figure 4, and Figure 5). On the one hand, when Tyr204 is far from the amino acid binding pocket (the open conformation), a π–π stacking interaction is observed between the imidazole ring of His225 and the indole ring of Trp237. On the other hand, when Tyr204 is located inside the amino acid binding pocket (the closed conformation), the imidazole ring of His225 shifts and is stabilized by a π–π stacking interaction with the aromatic ring of Tyr204. This conformational change is not observed in the corresponding His227 residue of *Ma*PylRS, according to the structure of the *Ma*PylRS apo form (Figure 4d,e, Figure 5 and Appendix A). However, the recently determined structure of the *Ma*PylRS mutant in complex with the non-canonical amino acid acridonylalanine (and AMPPNP) revealed the conformational changes of residues 224–230, and the movement of His227 away from the active site upon acridonylalanine binding (Figure 4f and Appendix A) [76]. In contrast, no conformational changes of the corresponding Ile405 residue in *Mm*PylRS are observed (Figure 4g,h and Appendix A). Accordingly, the structural changes of His225/His227 share common features with *Ma*PylRS and ISO4-G1 PylRS, but not with Ile405 of *Mm*PylRS.

### 2.6. Structure-Based Engineering of the First-Layer Residues in ISO4-G1 PylRS for Site-Specific Incorporation of Bulky Lysine Derivatives into Proteins by Cell-Free Protein Synthesis

Previously, we developed a system for genetic code expansion with bulky ZLys derivatives by using the *Mm*PylRS (Y306A/Y384F)·tRNA^Pyl^ pair [8,63,69,71] and the *Ma*PylRS (Y126A/M129L)·tRNA^Pyl^ pair [24,26]. To examine whether the ISO4-G1 PylRS·tRNA^Pyl^ pair is useful for genetic code expansion, we rationally engineered two ISO4-G1 PylRS mutants from the PylRS structures. In the previous study, the *Ma*PylRS (Y126A/M129L)·tRNA^Pyl^ and *Ma*PylRS (Y126A/M129A)·tRNA^Pyl^ pairs successfully facilitated the site-specific incorporation of TCO*Lys and *m*AzZLys into proteins in an *E. coli* cell-free protein synthesis system [26]. Therefore, the Y126A/M129L and Y126A/M129A mutations of the “first-layer residues”, which directly contact the substrate amino acids, were transplanted into the corresponding sites (Tyr125 and Met128) of ISO4-G1 PylRS. The Y125A mutation (corresponding to Y306A in *M. mazei* PylRS and Y126A in *M. alvus* PylRS) enlarges the ISO4-G1 PylRS active site pocket, which then becomes suitable for accommodating bulky non-canonical amino acids [26]. In ISO4-G1 PylRS, the Met128 side-chain protrudes into the amino acid binding pocket (Figure 3), which would reduce the pocket size as compared with that of *Mm*PylRSc. The Leu and Ala mutations at position 128 would enlarge the inner space of the active site pocket (Figure 3). All of the ISO4-G1 PylRS mutant proteins were quite soluble, and over 100 mg quantities of the purified ISO4-G1 PylRS proteins per liter *E. coli* culture were obtained (Materials and Methods).

Using the ISO4-G1 PylRS mutants, we tested cell-free protein synthesis for the site-specific incorporation of bulky non-canonical amino acids, such as ZLys, *m*AzZLys, *N*^ε^-(*p*-azidobenzyloxycarbonyl)-L-lysine (*p*AzZLys), *N*^ε^-(*p*-ethynylbenzyloxycarbonyl)-L-lysine (*p*EtZLys), and TCO*Lys (Figure 1), into the N11-GFPS1 protein in response to an amber (UAG) codon at position 17, using the cell extract of the RF-1 (*prfA*) deletion *E. coli* strain B-60.ΔA::Z [26,47]. The yields of the N11-GFPS1 proteins containing ZLys, TCO*Lys, *p*EtZLys, *m*AzZLys, and *p*AzZLys by using the ISO4-G1 PylRS (Y125A/M128A)·tRNA^Pyl^ pair were 2.8, 0.9, 0.3, 2.8, 0.9 mg protein/mL reaction, respectively, (132%, 41%, 15%, 131%, and 41% productivities, respectively, relative to the N11-GFPS1 control protein) (Figure 6a).

In contrast, with the ISO4-G1 PylRS (Y125A/M128L)·tRNA^Pyl^ pair, the yields of the N11-GFPS1 proteins containing ZLys, TCO*Lys, *p*EtZLys, *m*AzZLys, and *p*AzZLys were 2.7, 2.9, 0.3, 3, and 2.3 mg protein/mL reaction, respectively, (125%, 136%, 14%, 140%, and 106% productivities, respectively, relative to the N11-GFPS1 control protein) (Figure 6a). The ISO4-G1 PylRS (Y125A/M128L) mutant achieved more than 100% protein productivities for ZLys, *m*AzZLys, *p*AzZLys, and TCO*Lys (Figure 6a), while the ISO4-G1 PylRS (Y125A/M128A) did so only for ZLys and *m*AzZLys. The amino acid-binding pocket of the ISO4-G1 PylRS (Y125A/M128A) mutant might not be suitable for TCO*Lys and *p*AzZLys, unlike that of the ISO4-G1 PylRS (Y125A/M128L) mutant. Unfortunately, both ISO4-G1 PylRS mutants achieved only 14–15% of the protein productivity for *p*EtZLys, indicating that this substrate is too large for them. In contrast, the protein productivities for *p*EtZLys with the *Mm*PylRS (Y306A/Y384F)·tRNA^Pyl^, *Ma*PylRS (Y126A/M129L)·tRNA^Pyl^, and *Ma*PylRS (Y126A/M129L/H227I/Y228P)·tRNA^Pyl^ pairs were 0.1, 0.14, and 0.34 mg/mL reaction (5, 7, and 13% protein productivities, respectively, relative to the N11-GFPS1 control protein), which were comparable to or less than that of ISO4-G1 PylRS (Y125A/M128L) (Figure 6b). Consequently, the protein productivities of the ISO4-G1 PylRS (Y125A/M128L)·tRNA^Pyl^ pairs with TCO*Lys were drastically enhanced, and comparable to those obtained using the *Ma*PylRS (Y126A/M129L/H227I/Y228P)·tRNA^Pyl^ pair [26].

### 2.7. Effects of the ISO4-G1 PylRS(Y125A/M128L) Concentration on Cell-Free Protein Synthesis with the Inefficient Amino Acid pEtZLys

Previously, we found that the protein productivities for the inefficient, bulky amino acid TCO*Lys can be enhanced by increasing the concentration of the *M. alvus* PylRS mutant [26]. To examine the effects of higher concentrations of the ISO4-G1 PylRS protein on non-canonical amino acid incorporation, cell-free protein synthesis with the super-inefficient, bulky, non-canonical amino acid *p*EtZLys, which is useful for alkyne-azide click chemistry [63,82], was performed by using various concentrations of the ISO4-G1 PylRS protein. The protein productivities for *p*EtZLys were enhanced from 8% (0.19 mg protein/mL reaction) to 57% (1.32 mg protein/mL reaction) of the N11-GFPS1 control protein when the concentration of the ISO4-G1 PylRS (Y125A/M128L) protein was increased from 10 to 75 μM (Figure 6c). Therefore, we achieved the highest ever protein productivity for *p*EtZLys. The incorporations of the non-canonical amino acids into the N11-GFPS1 protein were confirmed by mass spectrometry analyses (Appendix A). These results confirmed that the efficient synthesis of the full-length N11-GFPS1 protein containing non-canonical amino acids occurs without any non-specific suppression of the UAG codon with canonical amino acids in the cell-free system.

### 2.8. Effects of the Second-Layer Mutations of ISO4-G1 PylRS for Site-Specific Incorporation of Bulky Lysine Derivatives into Proteins by Cell-Free Protein Synthesis

We found that the productivities of N11-GFPS1 proteins containing ZLys, *m*AzZLys, and TCO*Lys obtained with the ISO4-G1 PylRS (Y125A/M128L) system were comparable to or higher than those from the *Ma*PylRS (Y126A/M129L/H227I/H228P) system (Figure 6a) [26]. Previously, the second-layer IP (H227I/H228P) additional mutations of *Ma*PylRS (Y126A/M129L) extensively improved the protein productivities for *m*AzZLys and TCO*Lys [26]. However, mutations of the second-layer residues His227 and Tyr228 in *Ma*PylRS (corresponding to the Ile405 and Pro406 residues in *Mm*PylRSc, respectively, (Figure 4, Appendix A)) might affect the first-layer residues, which interact directly with substrate amino acids within the amino acid binding pocket [26].

Interestingly, the ISO4-G1 PylRS structure revealed that ISO4-G1 PylRS His225, corresponding to *Ma*PylRS His227, undergoes drastic conformational changes in accordance with the location of Tyr204 (corresponding to Tyr206 in *Ma*PylRS) in the β5-β6 hairpin (Figure 5 and Appendix A). We introduced the H225A mutation in ISO4-G1 PylRS and examined its effects on protein productivity. The ISO4-G1 PylRS (H225A) mutant had significantly decreased protein productivities for non-canonical amino acids (Figure 7). The yields of the N11-GFPS1 proteins containing *N*^ε^-(*t*-butyloxycarbonyl)-L-lysine (BocLys) and *N*^ε^-propargyloxycarbonyl-L-lysine (PocLys) (Figure 1) were only 0.096 and 0.048 mg protein/mL reaction, respectively, (4% and 2% productivities, respectively, of the N11-GFPS1 control protein) (Figure 7). These biochemical and crystallographic analyses confirmed that the His225 residue is crucial for the ISO4-G1 PylRS activity and cannot be replaced.

## 3. Discussion

In the present study, we determined the crystal structure of ISO4-G1 PylRS, and by its structure-based engineering, we achieved full productivity of cell-free protein synthesis according to the expanded genetic code with a variety of bulky non-canonical amino acids. By introducing two mutations into the first layer of the amino acid-binding pocket in ISO4-G1 PylRS, we achieved full productivity of cell-free synthesis with ZLys, TCO*Lys, *m*AzZLys, and *p*AzZLys. The first-layer mutant of ISO4-G1 PylRS required no additional second-layer mutations for the full productivity with these bulky non-canonical amino acids. Even with the much bulkier and most inefficient non-canonical amino acid, *p*EtZLys, we finally achieved the highest-ever levels of protein productivity by using the ISO4-G1 PylRS (Y125A/M128L) protein at a 7.5-fold higher concentration than the standard protocol. So far, this drastic improvement of protein productivity for *p*EtZLys has never been accomplished with the *M. mazei* and *M. alvus* systems.

Previously, we introduced the Y126A mutation of *Ma*PylRS (corresponding to the Y306A mutation of *Mm*PylRS), and the M129L or M129A mutation in the first-layer residues [26]. We found that simply transplanting the *Ma*PylRS (Y126A/M129L or Y126A/M129A) mutations into ISO4-G1 PylRS was appropriate for bulky non-canonical amino acids. The two ISO4-G1 PylRS mutants (Y125A/M128L and Y125A/M128A) with enlarged amino acid binding pockets achieved full productivity and showed much higher activities than those of *Mm*PylRS (Y306A/Y384F) for ZLys, *m*AzZLys, *p*AzZLys, and TCO*Lys (Figure 3a and Figure 6a). However, the full productivity level has not yet been achieved for more difficult non-canonical amino acids, such as *p*EtZLys. Because ISO4-G1 PylRS, as well as *Ma*PylRS, is highly water-soluble, ISO4-G1 PylRS mutants can be used in the cell-free reaction at much higher concentrations than that of the standard protocol. Consequently, the yield of the *p*EtZLys-incorporated protein reached 1.3 mg/mL per cell-free reaction (57% productivity level of the control protein synthesis) when the concentration of the ISO4-G1 PylRS (Y125A/M128L) protein was increased up to 75 μM (Figure 6c).

The higher catalytic activity of ISO4-G1 PylRS than that of *Ma*PylRS in the cell-free system was achieved for the site-specific incorporation of *N*^ɛ^-(2-(trimethylsilyl)ethoxycarbonyl)-L-lysine into proteins [28]. The molecular mechanism underlying this higher catalytic activity of ISO4-G1 PylRS than those of *Ma*PylRS and *Mm*PylRS remains unknown. Based on the crystal structure of ISO4-G1 PylRS (Figure 2, Figure 4 and Figure 5), the β5–β6 hairpin may exist in a dynamic open-closed equilibrium, and the location and conformational change of the His225 residue appear to be important for the catalytic activity. The ISO4-G1 PylRS His225 residue is conserved as His227 in *Ma*PylRS, and undergoes a drastic conformational change upon non-canonical amino acid (and AMPPNP)-binding (Figure 4 and Appendix A) [76]. However, in *Ma*PylRS, His227 does not interact with Tyr206 and Trp241, in contrast to the interactions of His225 with Tyr204 and Trp237 in ISO4-G1 PylRS. The ISO4-G1 PylRS mutant with His225 replaced by Ala225 abolished the protein productivities for non-canonical amino acids (Figure 7). In the case of ISO4-G1 PylRS, the His225Ala mutation might reduce the interactions of His225 with Tyr204 and Trp237. In the above-mentioned dynamic closed-open equilibrium of the hairpin, the degree of movement of the hairpin in ISO4-G1 PylRS may be comparable to those in *Ma*PylRS and *Mm*PylRS, concerning the tip positions between the open and closed forms; although, we still lack ISO4-G1 PylRS structures bound to amino acid substrates (Figure 2, Figure 3, Figure 4 and Appendix A). The interactions of His225 with Tyr204 and Trp237 in ISO4-G1 PylRS (Figure 5), which are not observed in *Ma*PylRS, appear to be a driving force for the rapid conformational changes of the β5-β6 hairpin. The elucidation of the molecular mechanism underlying the higher catalytic activities of the ISO4-G1 PylRS mutants based on the PylRS structures will lead to the development of a next-generation platform for producing non-canonical amino acid-incorporated proteins.

In the present study, we demonstrated that the ISO4-G1 PylRS system extensively improved the protein productivities, even for the very difficult, non-canonical amino acid *p*EtZLys, which had not been achieved by the *Mm*PylRS and *Ma*PylRS systems. The ISO4-G1 PylRS·tRNA^Pyl^ pair, rationally engineered based on the ISO4-G1 PylRS crystal structures, will serve as a more useful tool for next-generation genetic code expansion technologies.

## 4. Materials and Methods

### 4.1. Materials

Biochemical and molecular biological procedures were performed with commercially available materials, enzymes, and chemicals. ZLys was purchased from Bachem (Bubendorf, Switzerland). *m*AzZLys, *p*EtZLys, and *p*AzZLys, were purchased from Sundia, Shanghai, China). BocLys was purchased from Watanabe Chemical, Hiroshima, Japan). PocLys and TCO*Lys were purchased from SiChem GmbH, Bremen, Germany).

### 4.2. Bacterial Strains and Plasmids

The DNA fragment encoding PylRS (ISO4-G1 PylRS, residues 1–273) from the methanogenic archaeon ISO4-G1 was chemically synthesized (Integrated DNA Technologies), PCR-amplified, and cloned into the pET28c vector (Appendix A). The *E. coli* BL21-Gold (DE3) strain (Novagen) was used for the expression of PylRS proteins.

### 4.3. Expression and Purification of PylRS Proteins

The pET28c vector plasmids containing the ISO4-G1 PylRS gene were transformed into the *E. coli* BL21-Gold (DE3) strain, and selected on LB agar plates supplemented with 50 μg/mL kanamycin. A single colony was grown at 37 °C in broth culture, containing 15 g tryptone, 7.5 g yeast extract, and 15 g NaCl per liter, supplemented with 50 μg/mL kanamycin. Expression of the N-terminally hexahistidine-tagged ISO4-G1 PylRS protein was induced with 1 mM IPTG when the OD_600_ reached 0.6. The cultivation temperature was then lowered to 20 °C, and the culture was continued overnight. The *E. coli* cells were collected by centrifugation and stored at −80 °C. The cells were resuspended in 50 mM potassium phosphate buffer (pH 7.4), containing 500 mM NaCl, 25 mM imidazole, 5 mM β-mercaptoethanol, and protease inhibitor cocktail (Complete-EDTA free ULTRA, Roche, Basel, Switzerland) (buffer C), and were sonicated on ice. The cell lysate was centrifuged at 15,000× *g* for 15 min at 4 °C, and the supernatant was applied to a HisTrap column (Cytiva, Uppsala, Sweden), which was equilibrated with buffer C. The protein was eluted with buffer C containing 400 mM imidazole, instead of 25 mM imidazole, and peak fractions were collected. The protein fractions were pooled, concentrated, and applied to a HiLoad 16/60 Superdex 200 column (Cytiva, Uppsala, Sweden), and equilibrated with 30 mM potassium phosphate buffer (pH 7.4), containing 200 mM NaCl and 1 mM DTT. The eluted fraction was collected and dialyzed against 40 mM potassium phosphate buffer (pH 7.4), containing 50 mM NaCl and 1 mM DTT (buffer B). The histidine-tag peptide derived from the pET28c vector was cleaved with thrombin protease (1 u per 0.1 mg PylRS protein, Sigma-Aldrich, St. Louis, MO, USA) at 4 °C overnight. The dialyzed fraction was then loaded on a HiTrap Q column (Cytiva, Uppsala, Sweden), and after washing the column with buffer B, the bound proteins were eluted by a linear gradient of 50–635 mM NaCl. The eluted fractions were pooled, concentrated, and applied to a HiLoad 16/60 Superdex 200 column (Cytiva, Uppsala, Sweden), equilibrated with 30 mM potassium phosphate buffer (pH 7.4), containing 200 mM NaCl and 1 mM DTT. The eluted fractions were collected, dialyzed against 10 mM Tris-HCl buffer (pH 8.0), containing 150 mM NaCl, 10 mM MgCl2, and 10 mM β-mercaptoethanol, and concentrated by ultracentrifugation to 16.2 mg/mL. Aliquots of the ISO4-G1 PylRS protein were flash-cooled in liquid nitrogen and stored at −80 °C. The *Ma*PylRS and *Mm*PylRS proteins were purified as described previously [26]. The histidine-tagged PylRS proteins were purified by chromatography on HisTrap and Superdex 200 HiLoad columns. After dialysis, the eluted PylRS proteins were concentrated by ultracentrifugation.

### 4.4. Preparation of tRNA^Pyl^ Transcripts

The tRNA^Pyl^s from the methanogenic archaeon ISO4-G1, *M. alvus*, and *M. mazei* were transcribed in vitro with T7 RNA polymerase, using the PCR-amplified DNA fragment as the template. The tRNA transcripts were precipitated with isopropanol, applied onto a Resource Q column (Cytiva, Uppsala, Sweden) equilibrated with 10 mM Tris-HCl buffer (pH 7.5), containing 5 mM MgCl_2_ and 50 mM NaCl, and eluted by a linear gradient of 0.05–0.7 M NaCl. The purified tRNA^Pyl^ transcripts were precipitated with ethanol and dissolved in 10 mM Tris-HCl buffer (pH 7.5) containing 5 mM MgCl_2_.

### 4.5. Crystallization, Data Collection, and Structure Determination

All crystallization screenings were performed by the sitting-drop vapor-diffusion method, by mixing 0.2 μL of the ISO4-G1 PylRS protein solution with 0.2 μL of reservoir solution, using a Mosquito liquid handling robot (TTP Labtech, now SPT Labtech, Melbourn, Hertfordshire, UK). Crystals were grown at 20 °C in conditions with 100 mM HEPES-NaOH buffer (pH 7.2), 20% PEG3350, and 200 mM KCl. The crystal was transferred to 100 mM HEPES-NaOH buffer (pH 7.2) containing 20% PEG3350, 200 mM KCl, and 18% trehalose, mounted on a nylon loop, and flash-cooled in liquid nitrogen. The X-ray diffraction datasets were collected at the beamline BL32XU in SPring-8 (Harima, Japan) at −173 °C and were processed with XDS [83]. The crystal of ISO4-G1 PylRS belongs to the space group P2_1_2_1_2_1_, with unit cell parameters of a = 98.51 Å, b = 102.68 Å, c = 349.86 Å, and α = β = γ = 90°. The phase was calculated by the molecular replacement method with Phaser, using 6JP2 as the search model. Ten ISO4-G1 PylRS molecules were found per asymmetric unit, with a solvent content of 56.6%. Iterative cycles of model refinement by PHENIX [84] and manual model building with Coot [85] were performed. The *R*_work_ and *R*_free_ factors for the ISO4-G1 PylRS structure are shown in Appendix A. The final model was validated with Molprobity [78] and Procheck [77]. Graphical images were prepared with PyMOL (http://pymol.sourceforge.net/ (accessed on 28 May 2020).). The statistics of the data collection and refinement are summarized in Appendix A.

### 4.6. Cell-Free Protein Synthesis and Purification of GFP Proteins Containing Non-Canonical Amino Acids

Cell-free coupled transcription/translation was performed as described previously [35,36,38,86], using pCR2.1-TOPO bearing the gene encoding an N11-tagged superfold type green fluorescent mutant protein (N11-GFPS1) [87]. The tRNA^Pyl^ transcripts were prepared by in vitro transcription [38,52]. The PylRS proteins were overproduced in *E. coli* BL21-Gold (DE3) cells and purified as described previously [8]. The pCR2.1-N11-GFPS1 plasmids containing the wild-type N11-GFPS1 gene or the mutant with a single UAG codon at position Ala17 were used as the template DNAs for cell-free protein synthesis with S30 extracts from RF-1 (*prfA*) deletion *E. coli* B-60ΔA::Z cells [47], and with a pMINOR plasmid encoding rare codon tRNAs [88]. The reaction components for the incorporation of non-natural lysine derivatives at position 17 in N11-GFPS1 were as follows: 2 μg/mL template plasmid, 10 μM PylRS, 10 μM tRNA^Pyl^, and 1 mM non-natural lysine derivatives. After an overnight incubation at 25 °C, the synthesized full-length N11-GFPS1 proteins were quantified as described previously, using an ARVO Victor2 V Multilabel Counter plate fluorescence reader (PerkinElmer, Waltham, MA, USA) [38]. The N11-GFPS1 proteins were purified as follows. After centrifugation of the solution, the supernatant fractions were loaded on a Ni-Sepharose High Performance column (Cytiva, Uppsala, Sweden). The column was washed with 50 mM potassium phosphate buffer (pH 7.4), containing 500 mM NaCl, 25 mM imidazole, and 5 mM β-mercaptoethanol, and then eluted with 50 mM potassium phosphate buffer (pH 7.4), containing 500 mM NaCl, 400 mM imidazole, and 5 mM β-mercaptoethanol. The PMF analyses of the N11-GFPS1 proteins, containing ZLys, *m*AzZLys, *p*AzZLys, *p*EtZLys, and TCO*Lys, were performed as described above.

## 5. Patents

A PCT international patent application [WO2020/045656 A1] related to this work has been filed.

## Data Availability

The coordinates and structure factors have been deposited in the RSCB Protein Data Bank (ID code 8IFJ for ISO4-G1 PylRS).

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
