# Peer review of "Crystal Structure of Pyrrolysyl-tRNA Synthetase from a Methanogenic Archaeon ISO4-G1 and Its Structure-Based Engineering for Highly-Productive Cell-Free Genetic Code Expansion with Non-Canonical Amino Acids"

_ijms, 2023, doi:10.3390/ijms24076256_

Round 1

Reviewer 1 Report

Dear authors, dear editor,

Thank you for letting me review this manuscript. Here, Yanagisawa et al. describe a new aminoacyl synthetase for pyrrolysine analogs that achieves improved yields during in vitro genetic code expansion. The authors provide a good background of the new synthestase, discuss the discovered structure in detail in comparison with established synthetases and investigate the incorporation of several important ncAAs into test proteins in cell extract. They discover a combination of mutations and conditions that beat the previously best synthetases in terms of yield during in vitro translation.

The paper is well written and provides the necessary data and references. Advancements such as these are very important for the field of unnatural protein synthesis that still struggles with diminished yields compared to regular protein production. This publication will therefore be of great interest and the good documentation of methods and results should enable other researchers to improve their synthesis strategies based on this.

One criticism I have is that the study is limited to in vitro translation of a simple GFP control. It would have been great to see the performance of the new synthetase in E. coli or even eukaryotic cells, as well as the inclusion of a more complex protein. Also, the work stops at the “obvious” mutations of analogous amino acids (admittedly with great results) but does little to screen for potential further improvements.

Still, a new “best in class” synthetase for a highly relevant combination of ncAAs and applications is clearly important enough to warrant publication in this special issue of Molecular Sciences.

I would therefore recommend publishing this manuscript with a few minor corrections:

Figure 4: This one could use some graphics editing. Even fully zoomed in it is difficult to recognize anything. Maybe make the colored regions more transparent or use a different color scheme altogether.

Figure 5 and accompanying text passages: The combination of structures, minor and capital letters and associated colors is difficult to follow. I appreciate the comprehensive and systematic approach of explaining things in this section but maybe there is a way to label things that makes it easier to grasp on the first read.

General: At some points in the main text the narrative goes into too much detail. E.g. the ESI analysis in paragraph 381ff is important but could be relegated to the methods or supplement in order to streamline the reading experience.

Supplement: The amino acid sequence of the new synthetase is described in the supplement but it would be good if the plasmid sequences were also included. Or, alternatively, a link to a public repository like Addgene.

Citations: The paper does include a significant amount of self-citation. This is mostly justified since they improve on some of their previously published works, but the editor should make sure that competitors are also cited where appropriate.

Author Response

March 22, 2023

Ms. Queenie He

Assistant Editor

Dear Ms. He,

   Thank you for your e-mail of March 13, 2023, regarding our manuscript (Manuscript ID: ijms-2268100), which we submitted for publication in the International Journal of Molecular Sciences. We were pleased to have learned that our manuscript would be acceptable if we properly addressed the points raised by the Reviewers. We are grateful to the Reviewers for their constructive comments and have addressed each of their points in our revision. Thus, we are submitting herewith the revised version of our manuscript.

   The changes are explained one by one below. Each of the Reviewers’ comments is preceded by a greater-than sign and written in italics and is followed by our response in regular font.

>Reviewer#1 comment 1

Thank you for letting me review this manuscript. Here, Yanagisawa et al. describe a new aminoacyl synthetase for pyrrolysine analogs that achieves improved yields during in vitro genetic code expansion. The authors provide a good background of the new synthetase, discuss the discovered structure in detail in comparison with established synthetases and investigate the incorporation of several important ncAAs into test proteins in cell extract. They discover a combination of mutations and conditions that beat the previously best synthetases in terms of yield during in vitro translation. The paper is well-written and provides the necessary data and references. Advancements such as these are very important for the field of unnatural protein synthesis that still struggles with diminished yields compared to regular protein production. This publication will therefore be of great interest and the good documentation of methods and results should enable other researchers to improve their synthesis strategies based on this.

One criticism I have is that the study is limited to in vitro translation of a simple GFP control. It would have been great to see the performance of the new synthetase in E. coli or even eukaryotic cells, as well as the inclusion of a more complex protein. Also, the work stops at the “obvious” mutations of analogous amino acids (admittedly with great results) but does little to screen for potential further improvements. Still, a new “best in class” synthetase for a highly relevant combination of ncAAs and applications is clearly important enough to warrant publication in this special issue of Molecular Sciences.

I would therefore recommend publishing this manuscript with a few minor corrections:

Response

We would like to express our gratitude to Reviewer#1 for his/her thorough understanding of our work and helpful comments.

>Reviewer#1 comment 2

Figure 4: This one could use some graphics editing. Even fully zoomed in it is difficult to recognize anything. Maybe make the colored regions more transparent or use a different color scheme altogether.

Response

According to Reviewer #1’s comments, we have made the colored regions of Figure 4 more transparent.

>Reviewer#1 comment 3

Figure 5 and accompanying text passages: The combination of structures, minor and capital letters and associated colors is difficult to follow. I appreciate the comprehensive and systematic approach of explaining things in this section but maybe there is a way to label things that makes it easier to grasp on the first read.

Response

We have rearranged the panels of Figure 5 for clarity: The Fig.5 panels c-g represent molecules in which His225 interacts with Trp237, and the Fig.5 panels h-l represent molecules in which His225 interacts with Tyr204 (lines 264-269).

>Reviewer#1 comment 4

General: At some points in the main text the narrative goes into too much detail. E.g. the ESI analysis in paragraph 381ff is important but could be relegated to the methods or supplement in order to streamline the reading experience.

Response

We have moved the detailed description of MS analyses (the former lines 410-419) to the caption of Figure S6.

>Reviewer#1 comment 5

Supplement: The amino acid sequence of the new synthetase is described in the supplement but it would be good if the plasmid sequences were also included. Or, alternatively, a link to a public repository like Addgene.

Response

We have added the sequence of the expression plasmid containing the ISO4-G1 PylRS(Y125A/M128L) gene in Table S2.

>Reviewer#1 comment 6

Citations: The paper does include a significant amount of self-citation. This is mostly justified since they improve on some of their previously published works, but the editor should make sure that competitors are also cited where appropriate.

Response

We have checked and rearranged the references and added 12 citations, including 11 from competitors (refs. 29, 30, 31, 41, 42, 66, 67, 68, 69, 70, 72, 74).

We hope that these changes have now made our manuscript acceptable for publication in International Journal of Molecular Sciences.

All correspondence concerning this manuscript should be addressed to:

Dr. Shigeyuki Yokoyama

RIKEN Yokoyama Laboratory,

1-7-22 Suehiro-cho, Tsurumi, Yokohama 230-0045, Japan

Phone: +81-45-503-9196; Facsimile: +81-45-503-9195 

I would like to thank you in advance for your favorable consideration.

Sincerely,

Shigeyuki Yokoyama

Reviewer 2 Report

Pyrrolysyl-tRNA synthetase (PylRS) and tRNAPyl(CUA) pairs are gifts from mother nature which allows site-specific incorporation of unnatural amino acids into proteins. Recently, ISO4-G1 PylRS was showed to perform well in unnatural amino acids incorporation, and it is orthogonal to another widely-used pyrrolysyl-tRNA synthetase, MmPylRS. In the present study, the authors solved the crystal structure of the wild-type ISO4-G1 PylRS apo form and compared with that of the multiple mutants of ISO4-G1 PylRS, which is able to genetically code cyanopyridylalanine into proteins. The authors had demonstrated that His225 residue of ISO4-G1 PylRS is important for the catalytic activity, and achieved high protein productivities by rational design of ISO4-G1 PylRS, even for pEtZLys, which had not been achieved by the MmPylRS and MaPylRS systems. This manuscript provides guiding principles for engineering ISO4-G1 PylRS, and proved the importance of second-layer mutations (mutations that do not directly contact with substrate amino acid). I support its publication in international journal of molecular sciences.

Author Response

March 22, 2023

Ms. Queenie He

Assistant Editor

Dear Ms. He,

   Thank you for your e-mail of March 13, 2023, regarding our manuscript (Manuscript ID: ijms-2268100), which we submitted for publication in the International Journal of Molecular Sciences. We were pleased to have learned that our manuscript would be acceptable if we properly addressed the points raised by the Reviewers. We are grateful to the Reviewers for their constructive comments and have addressed each of their points in our revision. Thus, we are submitting herewith the revised version of our manuscript.

   The changes are explained one by one below. Each of the Reviewers’ comments is preceded by a greater-than sign and written in italics and is followed by our response in regular font.

>Reviewer#2 comment

Pyrrolysyl-tRNA synthetase (PylRS) and tRNAPyl pairs are gifts from mother nature which allows site-specific incorporation of unnatural amino acids into proteins. Recently, ISO4-G1 PylRS was showed to perform well in unnatural amino acids incorporation, and it is orthogonal to another widelyused pyrrolysyl-tRNA synthetase, MmPylRS. In the present study, the authors solved the crystal structure of the wild-type ISO4-G1 PylRS apo form and compared with that of the multiple mutants of ISO4-G1 PylRS, which is able to genetically code cyanopyridylalanine into proteins. The authors had demonstrated that His225 residue of ISO4-G1 PylRS is important for the catalytic activity, and achieved high protein productivities by rational design of ISO4-G1 PylRS, even for pEtZLys, which had not been achieved by the MmPylRS and MaPylRS systems. This manuscript provides guiding principles for engineering ISO4-G1 PylRS, and proved the importance of second-layer mutations (mutations that do not directly contact with substrate amino acid). I support its publication in international journal of molecular sciences.

Response

We would like to express our gratitude to Reviewer#2 for his/her thorough understanding of our work and helpful comments.

We hope that these changes have now made our manuscript acceptable for publication in International Journal of Molecular Sciences.

All correspondence concerning this manuscript should be addressed to:

Dr. Shigeyuki Yokoyama

RIKEN Yokoyama Laboratory,

1-7-22 Suehiro-cho, Tsurumi, Yokohama 230-0045, Japan

Phone: +81-45-503-9196; Facsimile: +81-45-503-9195 

I would like to thank you in advance for your favorable consideration.

Sincerely,

Shigeyuki Yokoyama

Reviewer 3 Report

This is a peer review of the article “Crystal structure of pyrrolysyl-tRNA synthetase from a methanogenic archaeon ISO4-G1 and its structure-based engineering for highly-productive cell-free genetic code expansion with non-canonical amino acids” submitted to the International Journal of Molecular Sciences by Yanagisawa and collaborators.

In this manuscript the authors focused on the pyrrolysyl-tRNA synthetase (PylRS) from the methanogenic archaeon ISO4-G1. They isolated the corresponding enzyme, solved its 3D structure, rationally engineered its active site to accommodate a series of large noncanonical amino acids, and assessed its potential for the translation of enhanced proteins through cell-free assays.

The study is well-rounded, and the manuscript is properly organized and illustrated. The PylRS methodically engineered by Yanagisawa and collaborator performs significantly better than benchmark synthetases in the field for the site-directed incorporation of highly relevant but notoriously recalcitrant synthetic amino acids such as TCO*Lys and pEtZLys. This work is very likely to grab the interest of many investigators in synthetic biology and inspire the design of the next generation of orthologous aminoacylation systems.

I recommend the publication of this manuscript in the International Journal of Molecular Sciences after the authors address the following minor comments:

Line 127: “excellent” is subjective – replace by “superior”

Page 9, Scheme 1: rank these non-natural amino acids by molecular weight or volume

Page 14: where is the His-tag on the recombinant PylRS? Is it in Nt or Ct?

Author Response

March 22, 2023

Ms. Queenie He

Assistant Editor

Dear Ms. He,

   Thank you for your e-mail of March 13, 2023, regarding our manuscript (Manuscript ID: ijms-2268100), which we submitted for publication in the International Journal of Molecular Sciences. We were pleased to have learned that our manuscript would be acceptable if we properly addressed the points raised by the Reviewers. We are grateful to the Reviewers for their constructive comments and have addressed each of their points in our revision. Thus, we are submitting herewith the revised version of our manuscript.

   The changes are explained one by one below. Each of the Reviewers’ comments is preceded by a greater-than sign and written in italics and is followed by our response in regular font.

>Reviewer #3 comment 1

This is a peer review of the article “Crystal structure of pyrrolysyl-tRNA synthetase from a methanogenic archaeon ISO4-G1 and its structure-based engineering for highly-productive cell-free genetic code expansion with noncanonical amino acids” submitted to the International Journal of Molecular Sciences by Yanagisawa and collaborators. In this manuscript the authors focused on the pyrrolysyl-tRNA synthetase (PylRS) from the methanogenic archaeon ISO4-G1. They isolated the corresponding enzyme, solved its 3D structure, rationally engineered its active site to accommodate a series of large noncanonical amino acids, and assessed its potential for the translation of enhanced proteins through cell-free assays. The study is well-rounded, and the manuscript is properly organized and illustrated. The PylRS methodically engineered by Yanagisawa and collaborator performs significantly better than benchmark synthetases in the field for the site-directed incorporation of highly relevant but notoriously recalcitrant synthetic amino acids such as TCO*Lys and pEtZLys. This work is very likely to grab the interest of many investigators in synthetic biology and inspire the design of the next generation of orthologous aminoacylation systems.

I recommend the publication of this manuscript in the International Journal of Molecular Sciences after the authors address the following minor comments:

Response

We would like to express our gratitude to Reviewer#3 for his/her thorough understanding of our work and helpful comments.

>Reviewer#3 comment 2

Line 127: “excellent” is subjective – replace by “superior”.

Response

We have replaced “excellent” with “superior” (line 130).

>Reviewer#3 comment 3

Page 9, Scheme 1: rank these non-natural amino acids by molecular weight or volume

Response

We have ranked the non-natural amino acids by molecular weight (Scheme 1, lines 351-354).

>Reviewer#3 comment 4

Page 14: where is the His-tag on the recombinant PylRS? Is it in Nt or Ct?

Response

We have added the description of “N-terminally hexahistidine-tagged ISO4-G1 PylRS” (line 526). Furthermore, we also added the sequence of the expression plasmid containing the ISO4-G1 PylRS gene in Table S2.

We hope that these changes have now made our manuscript acceptable for publication in International Journal of Molecular Sciences.

All correspondence concerning this manuscript should be addressed to:

Dr. Shigeyuki Yokoyama

RIKEN Yokoyama Laboratory,

1-7-22 Suehiro-cho, Tsurumi, Yokohama 230-0045, Japan

Phone: +81-45-503-9196; Facsimile: +81-45-503-9195 

I would like to thank you in advance for your favorable consideration.

Sincerely,

Shigeyuki Yokoyama